# Advances in Mass Spectrometry on Membrane Proteins

**DOI:** 10.3390/membranes13050457

**Published:** 2023-04-24

**Authors:** Hsin-Chieh Yang, Weikai Li, Jie Sun, Michael L. Gross

**Affiliations:** 1Department of Chemistry, Washington University in St. Louis, St. Louis, MO 63130, USA; 2Department of Biochemistry and Molecular Biophysics, Washington University School of Medicine, St. Louis, MO 63110, USA; weikai@wustl.edu

**Keywords:** membrane protein, hydrogen–deuterium exchange, footprinting, fast photochemical oxidation of proteins (FPOP), mass spectrometry

## Abstract

Understanding the higher-order structure of membrane proteins (MPs), which are vital for numerous biological processes, is crucial for comprehending their function. Although several biophysical approaches have been used to study the structure of MPs, limitations exist owing to the proteins’ dynamic nature and heterogeneity. Mass spectrometry (MS) is emerging as a powerful tool for investigating membrane protein structure and dynamics. Studying MPs using MS, however, must meet several challenges including the lack of stability and solubility of MPs, the complexity of the protein–membrane system, and the difficulty of digestion and detection. To meet these challenges, recent advances in MS have engendered opportunities in resolving the dynamics and structures of MP. This article reviews achievements over the past few years that enable the study of MPs by MS. We first introduce recent advances in hydrogen deuterium exchange and native mass spectrometry for MPs and then focus on those footprinting methods that report on protein structure.

## 1. Introduction

Membrane proteins (MPs) are essential for a wide range of biological processes, including cell signaling, transport, and energy conversion, as well as for mediating interactions between the internal and external environments of the cell [1]. To understand the function of MPs, it is crucial to determine their higher-order structure. Biophysical techniques such as X-ray crystallography [2], nuclear magnetic resonance (NMR) spectroscopy [3], single molecule tracking with fluorescence spectroscopy [4], and cryo-electron microscopy (cryo-EM) [5] are being widely used to study the structure of MPs. These methods, however, have limitations in studying the dynamic nature and in dealing with the heterogeneity of MPs. The techniques of cryo-electron microscopy (cryo-EM) and X-ray crystallography can reveal the conformational state(s) of a protein. Cryo-EM has made significant strides in its improvement in resolution. Thus, cryo-EM can now provide structural information for MPs in nanodiscs, whereas X-ray crystallography still faces obstacles in obtaining high-quality crystals of MPs. However, directly probing protein dynamics remains a difficult task for both Cryo-EM and X-ray crystallography.

Mass spectrometry (MS) is emerging as a powerful tool for investigating localized structures and dynamics of MPs. Top-down and bottom-up approaches provide detailed information on protein primary structure, post-translational modifications, and protein–protein interactions on the “peptide” level and sometimes on the residue level. Recently, MS-based approaches are being used for interrogating MPs. MS now can provide information by examining the levels of protein organization, making use of structural proteomics [6]. Structural proteomics approaches include hydrogen/deuterium exchange (HDX) [7], chemical crosslinking (XL) [8], and footprinting (e.g., chemical labeling (CL), and hydroxyl radical footprinting (HRFP)). Various strategies for generating hydroxyl radicals, such as synchrotron radiolysis [9], fast photochemical oxidation of proteins (FPOP) [10], and plasma-induced modification of biomolecules (PLIMB) [11] have been developed. Downard [12] developed the free radical footprinting technique in an ESI source, which has been recently reviewed [13]. Additionally, Sharp et al. [14] present a new integrated FOX (Flash OXidation) Protein Footprinting System. Moreover, native MS has emerged as a new field for studying the stoichiometry of MPs subunits in their gas state and their interactions with other proteins and lipids [15].

The MS-based approaches used to study MPs have often been adapted from studies of water-soluble proteins. However, these methods must be refined to overcome challenges that are unique to MPs, and there are several. (1) Stability and solubility: MPs are often sensitive to changes in pH, temperature, and detergents, inducing denaturation or aggregation. Thus, it is difficult to maintain their stability and solubility during sample preparation and analysis. (2) Complexity of the protein–membrane system: MPs are embedded in a lipid bilayer, and their interactions with the surrounding lipids can play a crucial role in their function. Structural proteomics MS methods, however, require the protein to be isolated from the membrane and separated from the lipids to avoid contamination of the analytical HPLC column and the mass spectrometer. (3) Digestion and detection: owing to the lack of sites that can be recognized by common digestion enzymes and the hydrophobicity of the protein, MPs are hard to digest, leading to poor coverage of the protein. (4) Residue reactivity for footprinting: the presence of the lipid bilayer and/or the detergents surrounding the protein can affect the efficiency of footprinting. In addition, the residues buried in the bilayer are often less reactive.

Recently, native MS has emerged as a cutting-edge approach for investigating the higher-order structure of MPs [16]. Soft electrospray ionization can maintain non-covalent interactions, making it useful for MPs in their native environment [17,18]. Native-MS can also be used to study the effects of different lipids on the stability and dynamics of the protein. This approach has been advanced by Robinson and her colleagues [19], who have developed a sophisticated top-down native MS approach. A good example of their approach is an investigation of the pathway of rhodopsin signaling and regeneration, providing a potential model for G protein-coupled receptor (GPCR) drug discovery in native membrane environments [20]. For recent progress of native MS for MPs, we refer the reader to an informative review by Marty’s group [17].

In addition to native MS, hydrogen/deuterium exchange MS (HDX-MS) has also made significant progress in the resolution of MP dynamics, as detailed in several published articles [15,21,22,23]. Unlike native MS, HDX-MS involves incubating the MP in a deuterated solvent (D_2_O) to exchange the variable hydrogens of the protein backbone with deuterium. The mass uptake at different peptide residues are then analyzed by using MS. The method reveals structural dynamics by comparing the deuterium uptake of the same protein under several conditions. A difficulty is that water or D_2_O may not be of high concentration in the membrane.

This review aimed to highlight the use of some structural proteomics methods for studying MPs over the past years. Although MS-based approaches have been widely discussed [24], this review focuses on progress based on three categories of MPs: membrane-associated proteins, extra-membrane domains, and transmembrane domains of integral membrane proteins.

## 2. HDX-MS

### 2.1. Method Development

HDX-MS has been used for decades to characterize soluble proteins, and recently it has gained momentum for focusing on MPs and their dynamics [7]. HDX-MS involves labeling proteins with deuterium and then using MS to measure the extent of exchange of deuterium for hydrogen atoms in the protein as a function of time. This can provide information about the stability of different regions of the protein and show how those regions interact with the surrounding lipid environment. The application of HDX-MS to MPs began in the early 2000s. One of the first studies was of the G-protein-coupled receptors (GPCRs) that play a role in cellular signal transduction. Zhang et al. [25] published the first study in this field in 2010. The authors used a detergent to solubilize a 2-adrenergic GPCR and optimized the quantity of detergent, the composition of the quenching solution, and other important parameters of the LC steps. Since then, HDX MS has been used to study a wide range of MPs, including more examples of G protein-coupled receptors, ion channels, and transporters.

It is important, however, to know the lipid-protein interactions at the molecular level for understanding the conformational changes of MPs. One technical bottleneck in the HDX-MS experiments is the low sequence coverage that is caused by the scarcity of cleavage sites, the resistance to digested enzymes, and poor chromatographic separation. The Rand group [26] compared the digestion of four integral MPs, all transporters including a Cl-/H+ exchange transporter (CIC-ecl), a leucine transporter (LeuT), a dopamine transporter (DAT), and a serotonin transporter (SERT). Porcine pepsin and three alternative aspartic proteases were used either in-solution or as immobilized enzymes on-column to optimize the processing. Pepsin was the most productive for the digestion of ClC-ec1 and LeuT, providing coverage of 82.2 and 33.2% of the protein, whereas the alternative proteases were better than pepsin for the digestion of DAT and SERT. On the other hand, using urea instead of guanidine hydrochloride as a denaturant turns out to be beneficial for improving sequence coverage for MPs.

The presence of lipids, protein ligands, and reducing agents in samples often poses a challenge in HDX-MS analysis of membrane proteins and large protein assemblies. Calvaresi et al. [27] introduced a technique for eliminating undesired components from the HDX sample before conducting chromatographic separation and MS analysis. This method involves utilizing a compact size-exclusion chromatography (SEC) column that is incorporated with a standard HDX-MS setup, which is temperature-controlled. By utilizing this approach, the investigators found they could effectively eliminate lipid constituents from protein–lipid complexes, separate an antibody from an antigen during epitope mapping, and eliminate compounds that interfere with MS analysis during HDX-MS. The integration of the compact SEC column into the conventional HDX-MS setup is a simple process and also has the potential to be widely applicable in the HDX-MS analysis of challenging protein structures.

### 2.2. Applications

#### HDX Transmembrane Domains

Membrane transporters not only play a role in transporting poorly permeable solutes into the cell but in targeting drugs. A timely review illustrates the application of HDX-MS to secondary active transporters [28].

Although X-ray crystallography and high-resolution cryogenic EM can supply a static snapshot of the different states, the entire processing cannot be monitored. HDX-MS can reveal the structural dynamics of MPs with molecule-level resolution under native conditions without chemical labeling, and even with limited amounts of protein. HDX provides the ability to resolve structure-dynamic landscapes of MPs in their unbound and ligand-bound forms.

Politis’s group systematically investigated the conformational landscape of three representative transporters including xylose transporter (XylE), lactose permease (LacY), and glycerol-3-phosphate antiporter (GlpT). LacY and XylE are symporters. These transporter proteins are from *Escherichia coli* [22]. The investigators measured the difference in deuterium uptake (ΔHDX) between the mutants LacY G46W, XylE G58W, and GlpT G66W, and the wild-type (WT) transporter in detergent micelles. They determined that the three mutants have a higher uptake of deuterium on the extracellular side compared to the wild type when comparing the ΔHDX of the peptides. Conversely, the investigators observed that the intracellular side is relatively shielded from deuterium exchange.

By combining MD simulations and results from HDX-MS experiments, the conformational equilibrium between the outward-facing (OF) and inward-facing (IF) states of XylE and LacY, embedded in nanodiscs with several lipid compositions, can be modulated by phosphatidylethanolamine (PE) through its interactions with charged residue networks. In this work, The researchers developed a model of secondary transport that not only accounts for intracellular interactions but also incorporates the influence of conserved charge networks at the interface between lipids and proteins (Figure 1) [29].

Traditional structural approaches are limited in characterizing the dynamical ensembles of membrane proteins, whereas HDX-MS has emerged as a powerful tool to study their conformational dynamics, providing equilibrium information about relevant populations. While peptide-level exchange analysis is often used in conjunction with molecular simulations to gain a qualitative understanding of protein flexibility, HDX-MS methods affording higher spatial resolution hold promise for revealing atomistic details of the entire spectrum of conformational states that underlie protein function. Jia et al. [30] addressed an integrative strategy combining HDX-MS and ensemble modeling, benchmarked on XylE wild-type and mutant conformers, and applied it to different lipid environments and ligand-bound ensembles to uncover protein–ligand interactions in atomic detail. Through integrative HDX-MS modeling, this study showcases the potential to effectively quantify and visualize co-populated states of membrane proteins in the presence of diverse substrates and inhibitors.

It has been challenging to study full-length membrane proteins in lipid bilayers owing to the scarcity of automated methods and the negative effects of membrane lipids on chromatography and mass spectrometry. Anderson et al. [31] described a new workflow that enables fully automated HDX-MS analysis of full-length transmembrane proteins in lipid bilayers by depleting phospholipids using zirconium oxide beads and syringeless nanofilters. The method was successfully demonstrated using the single-pass transmembrane protein FcγRIIa, which showed optimal liquid chromatography-mass spectrometry performance and suitable amino acid sequence coverage needed for future measurements of structural dynamics. Moreover, Hammerschmid et al. [32] presented an extended HDX-MS system that automates the delipidation process of lipid-solubilized membrane proteins. An HDX-MS equipment was enhanced with the integration of a chromatographic phospholipid trap column, which enabled the online delipidation of samples before protease digestion of the deuterium-labeled protein–lipid assemblies. The setup allows proteins to pass through and undergo digestion with subsequent peptide trapping while retaining phospholipids in the ZrO_2_ matrix of the phospholipid trap column. The effectiveness and automation of phospholipid capture were successfully demonstrated on both empty and AcrB-loaded membrane scaffold protein–lipid nanodiscs, with minimal D-to-H back-exchange, peptide carry-over, and protein loss. The method can significantly overcome the challenges of membrane protein analysis and allow for better interrogation of their dynamics in artificial lipid bilayers or even native cell membranes.

As we know, HDX-MS can provide conformational information about membrane proteins, but HDX analysis on reconstituted in-vitro systems cannot represent the in-vivo environment. Donnarumma et al. [33] used outer-membrane vesicles naturally released by *Escherichia coli* to analyze native OmpF through HDX-MS, and a new protocol was developed to avoid interference from lipid contents. The extent of deuterium incorporation is consistent with the X-ray diffraction data, with buried β-barrels incorporating a low amount of deuterium and internal/external loops incorporating a higher amount. The kinetics of incorporation showed that peptides were segregated into two distinct groups based on trimeric organization, with fast-labeled peptides facing the surrounding environment and slow-labeled peptides located in the buried core. The study demonstrates that HDX-MS can address solvent accessibility and spatial arrangement of an integral outer membrane protein complex in a complex biological system.

### 2.3. Future Directions for HDX

The outlook for HDX-MS on MPs is promising. Advances have led to novel insights into the dynamic behavior of MPs, allowing the study of structural changes under difficult conditions. The structural-resolution capabilities of HDX-MS make it attractive for structural biology studies, as it can provide residue-level information about the protein. There are, however, challenges associated with the complexity of the MP. The hydrophobic parts of the MPs are challenges for the LC separation performance. Furthermore, the development of experimental and computational methods needs to be accelerated to shorten the gap between obtaining static snapshots from X-ray crystallography or CryoEM and determining the underlying conformational landscapes. The automated method for phospholipid removal described in this review may have significant implications for the structural characterization of membrane proteins and the development of pharmaceuticals targeting them. HDX can also be adapted for other MS applications such as protein enrichment for proteomics of extracellular vesicles [34]. The adaptable nature of HDX suggests its potential for various applications, and its relatively easy adoption makes it a promising tool for future studies of membrane proteins.

## 3. Chemical Footprinting

### 3.1. Method Development

Protein footprinting coupled with MS has proven successful in studying the higher-order structure of proteins, including in recent years those of MPs. As stated earlier, MPs are a challenging class of proteins owing to their hydrophobic nature and the inevitable presence of contaminants such as detergents and lipids in solutions prepared for the mass spectrometer. MS-based irreversible footprinting methods, when adjusted to overcome these challenges, can provide information about the dynamic behavior of MPs and their interactions. An advantage of irreversible labeling in footprinting is that it produces a stable, irreversible modification that can withstand the purification process employed in proteomic workflows, making the analysis more complicated but less vulnerable to the reversibility of HDX. It can provide a reliable and robust analysis of protein structure and dynamics, as the labeling modification remains intact throughout the entire process of sample preparation, purification, and analysis.

MS-based footprinting methods include chemical labeling (CL) [35,36,37] and hydroxy radical footprinting (HRF) including synchrotron radiolysis [9], fast photochemical oxidation of proteins (FPOP), or other methods mentioned earlier. Oxidative labeling is commonly used for analyzing the conformation changes of MPs, whereas FPOP, synchrotron HRF, and Fenton chemistry use different means of producing reactive oxygen species to label the protein. Other chemical footprinting reagents, such as carbenes, iodide radicals, and carbocations, are also reactive, introducing labels in the protein faster than protein unfolding [38]. Residue-specific reagents, such as diethylpyrocarbonate, glycine ethyl ester, and benzyl hydrazide, are more targeted methods that can provide information about specific residues of the protein, but these reagents require validation that the footprinting does not perturb structure.

MS-based footprinting is a valuable tool for studying the interactions of MPs with other molecules and their conformational changes. Some reviews discuss covalent labeling coupled with MS [39] and the fast footprinting of MPs [40]. In this review, we focus on several advanced footprinting methods coupled with MS for (i) membrane-associated proteins, (ii) extramembrane domains, and (iii) transmembrane domains, showing how methods can be developed to probe various domains of the protein.

### 3.2. Applications

#### 3.2.1. Footprinting Membrane-Associated Proteins

Membrane-associated proteins are physically attached to cellular membranes, either by covalent bonds or by non-covalent interactions [41,42]. These proteins play important roles in maintaining membrane structure, regulating cell signaling and transport, and facilitating communication between cells. An example of footprinting this type of MP was discussed by Van et al., in 2020 [43]. They investigated a membrane-distal conformation of the small GTPase KRAS by using a combination of neutron reflectivity, FPOP, and NMR. KRAS is positioned on the plasma membrane, where it plays a critical role in linking extracellular growth factor stimulation to intracellular signaling pathways (Figure 2). Defining the membrane-bound state of KRAS and gaining insight into the mechanism of the signal transfer are crucial for understanding its biological functions. The study focused on the membrane-associated protein alone or in a 1:1 ratio with nanodiscs. This is a compelling illustration that FPOP can interrogate the interactions of MPs with nanodiscs, and the outcome can inform on native membranes.

An earlier example shows the use of residue-specific reagents glycine ethyl ester (GEE) by Blankenship, Gross, and co-workers [44] for labeling solvent-accessible carboxyl groups on glutamic (E) and aspartic acids (D). The membrane-attached Fenna-Matthews-Olson (FMO) antenna protein plays a role in photosynthesis to absorb light and connect the large peripheral chlorosome antenna complex with the reaction center. The investigators compare three states: the solvent-exposed surfaces of isolated FMO protein, the FMO from chlorosome-depleted membranes, and the FMO from the native membrane. The reagent is covalently attached to glutamic acid (E) and aspartic acid (D) relatively rapidly in the presence of a mediator, water-soluble carbodiimide, 1-ethyl-3-(3-dimethylaminopropyl)carbodiimide (EDC). EDC is often used as a cross-linking reagent to activate the carboxyl groups of proteins. Once activated, the nucleophilic modifying reagent, GEE, reacts with the activated carboxyl group to produce the desired product [45]. The labeling sites were located and quantified by MS after protein purification and enzyme digestion. The modification levels of different peptides were compared to determine the interaction interfaces. The approach combines carboxyl group modification with MS to afford surface mapping or footprinting [46,47] of the protein and to uncover the interaction of a protein associated with membranes.

Recently, Vachet’s group [48] demonstrated that diethylpyrocarbonate (DEPC) can efficiently label specific amino acid residues to characterize the binding interactions between a membrane-associated protein and its binding partners (Figure 3). Using chemotaxis histidine kinase (CheA) as a model system, the investigators showed how DEPC-based covalent labeling can provide structural and binding information. Surprisingly, DEPC-based CL-MS is suitable for studying the interactions of membrane-associated proteins in artificial membranes even though DEPC has moderate hydrophobicity.

#### 3.2.2. Footprinting Extramembrane Domains

The capability to use MS to study MPs has emerged because it has high sensitivity, good structural resolution, relatively high throughput, and the capability to study proteins in complex mixtures [24]. MP structures, however, are difficult to resolve compared to structures of water-soluble proteins owing to the complexity of the assemblies [49]. The usual way to solve and stabilize an MP is by using detergents. But detergents affect protein conformation and hinder protein interactions with other molecules and reagents that footprint the protein. To overcome these problems, investigators reconstituted MPs in liposomes and bicelles that are chosen to mimic a native environment. Yue and co-workers [49] used Nanodiscs [50] to stabilize a light-harvesting complex 2 (LH2) from *Rhodobacter (Rb.) sphaeroides* in an aqueous buffer prior to and during labeling by hydroxyl radicals. They demonstrated MS-based FPOP footprinting of an MP complex in a near-native environment. The results show the protein’s outer membrane regions are footprinted more by •OH radicals than the regions spanning the lipid bilayer, which remain largely inert to the labeling.

Furthermore, Zhou and co-workers [51] showed for the first time the application of a picodisc system for MS footprinting of MPs. A novel methodology was used to incorporate an MP into saposin A picodiscs for MS footprinting. The saposin–lipoprotein picodisc [52,53] allows for the reconstitution of MPs in a lipid environment that is stabilized by a scaffold of saposin proteins. It is an advantageous methodology that provides a more native-like environment for MPs. The investigators achieved broad coverage that enables the analysis of the ferroportin structure. The picodisc system allows the protein to maintain its native folding topology during the footprinting, and the FPOP labeling occurs preferentially at the extramembrane regions of ferroportin.

*N-*Ethylmaleimide (NEM) also can be used in specific amino acid footprinting of Cys residues [54] of ferroportin in picodiscs. As usual, the investigators used proteomics-based MS analyses to locate the labeled sites. The investigators demonstrated that ferroportin, a membrane protein, is less stable in detergent micelles. The high sequence coverage could be achieved after reconstituting in picodiscs for MS footprinting. Although the FPOP labeling occurs preferably at the extramembrane regions of ferroportin, the NEM footprinting occurs in both extramembrane and intramembrane regions.

There are also other reactive species that can be chosen. One is the CF_3_ radical, which can label almost all amino residues [55,56]. It is a complement to the hydroxyl radical for addressing conformational changes. It can modify 18 of the 20 common amino acid residues, and it can be employed to investigate MPs. By using this reagent on vitamin K epoxide reductase (VKOR), the authors found that •OH was not reactive. Only ten residues on the extra-membrane regions were modified and some of them were Ala and Gly.

Protein conformation can be captured in its native state by footprinting with reactive carbocations possessing lifetimes as short as nanoseconds. Sun et al. [57] introduced carbocations (R_3_C^+^) as laser-initiated footprinting reagents for MPs. A trifluorobenzyl bromide (TFBB) reagent was tested for footprinting VKOR at pH 7.4. Most of the footprinting occurs on the extra-membrane region, cytosolic or extra-cytosolic (Figure 4). Moreover, amphiphilic TFB+ may offer the potential to footprint the transmembrane regions of integral MPs as confirmed by the discovery that TFB-modified peptides were observed in the transmembrane region, although the labeling efficiency was not high compared with residues in the solvent-accessible domains.

Some chemical labeling approaches modify specific amino acid side chains in slow reactions. These always have the disadvantage that slow labeling can be excessive and misleading because the protein undergoes conformational opening during the footprinting itself. Here the investigators choose a labeling strategy for specific amino acid side chains [58] that are hypothesized to be involved in the interactions. Schmidt et al. [59] used diethylpyrocarbonate (DEPC) to modify histidine residues. DEPC modifies lysine, arginine, tyrosine, cysteine, and threonine residues with different reactivities. The investigators also combined the MS approach with computational methods to improve the prediction of multiprotein complexes (i.e., F-type ATP synthase from spinach chloroplasts (cATPase)). The strategy helps investigators understand the conformational states of the peripheral stalk and allows localization of the flexible regions of the enzyme.

In another example, Zhou et al. [60] demonstrated a strategy to monitor ligand modulation of protein receptors. Protein complexes often have lysine residues located in or near the binding sites of small-molecule ligands. These lysine residues are crucial for facilitating protein–ligand interactions and can play a significant role in the binding and recognition of the ligand by the protein. The investigators used a dimethyl label to assess the solvent accessibility of lysine residues in catechol-*O*-methyltransferase and the *N*-methyl-*D*-aspartate receptors. Sixty-three lysine residues were comprehensively monitored. The results show that 20 lysine residues are involved in the ligand-binding and conformation-changing regions. The approach may provide an unbiased ligand modulation predicting strategy that can “scout” ligand-protein interactions.

#### 3.2.3. Footprinting Transmembrane Domains

Transmembrane regions of proteins span the cell membrane and have both intracellular and extracellular portions, which play important roles in regulating the transport of ions and small molecules across the membrane and in transmitting signals from the extracellular environment to the cell interior [61]. Examples of transmembrane proteins include ion channels, transporters, and receptors. Compared with water-soluble proteins, their higher-order structures and binding interactions are difficult to characterize owing to their partially hydrophobic surfaces and instability [62]. As mentioned above, Zhou and co-workers [51] showed that FPOP labeling occurs preferably at the extramembrane regions of ferroportin. NEM labels not only extramembrane but intramembrane regions. Because *N*-ethylmaleimide (NEM) is amphiphilic, it footprints cysteine residues in both extramembrane and transmembrane regions, thereby affording complementary footprinting coverage. Earlier, using •OH radicals produced by radiolysis, Bricker’s group [63] modified buried amino acid residues and demonstrated that water and oxygen channels are crucial for understanding the function of photosystems.

In addition to hydroxyl radicals, new reagents that are compatible with the FPOP platform are under development. One is the carbene diradical (:CR_2_), which can be used to study protein–protein interactions. A suitable precursor molecule is chosen to generate carbene radicals upon UV irradiation. Manzi et al. [35] demonstrated the use of photoactivatable aryl diazirines to create reactive carbenes that can be used to map the transmembrane region of the MP and ompF proteins from *E. coli*. The aryl diazirine precursor, due to its amphiphilic nature, can insert into micelles and generate carbenes upon laser irradiation. This allows for the mapping of hydrophobic transmembrane regions. Another means of generating carbenes is the FPOP platform, as demonstrated by Zhang, et al. [64]. Although carbenes have not been used for MPs on this platform, they offer the advantage that the precursor can be designed to be hydrophobic and readily partitioned to the membrane to be activated there. Furthermore, there are no reactive secondary products (e.g., radicals) produced in the footprinting as there must be with free radicals.

Another novel approach takes FPOP to a new level to offer a high coverage of the hydrophobic transmembrane (TM) regions. The provided resolution is enough to address structural inquiries with accuracy [65]. The method, developed by Sun et al. [65] and named NanoPOMP, can overcome some obstacles of MP footprinting. The investigators designed an approach to afford increased footprinting coverage of the TM region of an integral MP. To achieve higher coverage of the TM region, photocatalytic titanium dioxide (TiO_2_) nanoparticles are attached to the surface of the liposome and, upon irradiation by the same laser used in FPOP, generate high local concentrations of free radicals (including •OH). At the same time, the laser irradiation initiates a Paterno-Buchi reaction of the phospholipids comprising the liposome. That reaction is a photochemically allowed [2 + 2] cycloaddition of a lipid double bond with the carbonyl bond of acetone, perturbing the lipid layer. The reactions help radicals easily penetrate and footprint VKOR and hGLUT1 (human glucose transporter family 1). The results show good coverage of transmembrane helices and help to locate both the ligand-binding residues and the ligand-induced conformational changes in a transporter.

In another approach whereby the precursor of a reactive species is partitioned to the membrane for subsequent activation, Cheng et al. [66] used highly hydrophobic perfluoroisopropyl iodide (PFIPI) as the precursor to footprint both the hydrophobic intramembrane and the hydrophilic extramembrane domains of the IMP vitamin K epoxide reductase (VKOR). A key step in the protocol is the implementation of tip sonication to ensure penetration of PFIPI into the micelle interior. The footprinting is initiated by a laser pulse that presumably photolyzes C_3_F_7_—I→·C_3_F_7_ + ·I. The heptafluoro isopropyl radical reacts with side-chain Hs to give a protein-centered radical that is “capped” by the more stable iodine radical. The reaction is fast and has 100% coverage for Tyr and Trp (it may also react with histidine). The incorporation of the reagent with sonication does not perturb VKOR’s high-order structure as determined by checking its activity as an enzyme. By taking advantage of the high *logP* (Figure 5), small size, and suitability to form radicals upon 248 nm laser photolysis on the FPOP platform, the approach may be generally suitable for interrogating MPs.

Building on this work, Guo et al. [67] tested the hydrophobic reagent diethylpyrocarbonate (DEPC) for MP footprinting. This reagent, mentioned earlier, is specific for nucleophilic residues Lys, His, Tyr, Ser, Thr, Cys, and the N-terminus, which can map the VKOR structure without affecting protein structure. Although DEPC predominantly footprints the extramembrane domain, the labeling yield was increased by tip sonication to enhance reagent diffusion into the micelle medium. This method resulted in a total of 30 modified residues (including Lys, His, Tyr, Ser, and Thr), including nine residues in the TM domain. Whereas DEPC labeling without tip sonication footprinted 10 residues, only two residues (i.e., K41 and K158) were located in the TM domain (Figure 6). Overall, the combination of choosing a suitable reagent and employing tip sonication has promise for transmembrane domain footprinting.

## 4. Summary and Perspectives

The past decade has seen substantial advances in MS-based approaches for the structural analysis of MPs. These approaches, including native MS, HDX-MS, and MS-based footprinting, each have their own benefits and limitations. In native MS, there have been remarkable improvements in probing MP-lipid interactions. In footprinting, both HDX-MS and molecular modeling have added to our understanding of membrane protein function. HDX-MS has enabled the examination of whole MPs under varying conditions, presenting new opportunities to investigate membrane-protein interactions, substrate recognition, and transport-related conformational transitions. MS-based footprinting is also beginning to play a part in determining the structure, dynamics, and ligand interactions of MPs. Both approaches require attention to proper handling of MPs and to developing optimal isolation and MS analysis methods. The media in which the MP is placed, including detergent or artificial membranes, and the MS experimental approach play crucial roles. Choosing labeling reagents or a combination of them is vital to ensure high precision and coverage in analyzing MPs. Integrating MS data with that from other structural methods, such as cryo-EM and NMR, may expedite the study of complex systems by using an integrated structural biology approach. The integration of MS data into structural models can be challenging, especially for dynamic systems, but advances are being made. Exploring the interaction of lipids, small molecules as drugs, and other proteins with MPs, both in vitro and ultimately in vivo, is expected to see considerable progress in the coming years [68].

## Figures and Tables

**Figure 1 membranes-13-00457-f001:**
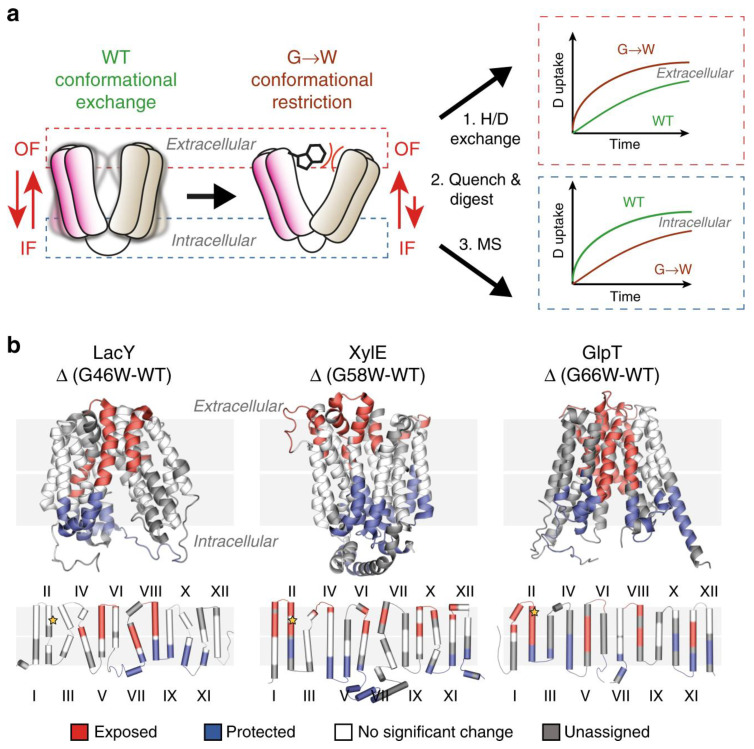
Use of G-to-W mutants in HDX-MS to examine changes in a conformational equilibrium between IF and OF states. (**a**) A comparison of deuteration levels between the WT and the mutant shows that intracellular peptides are more deuterated in the mutant, whereas the opposite is observed in the WT. (**b**) The topological mapping of mutated LacY, XylE and GlpT based on differential deuterium uptake (ΔHDX). Reproduced with permission from Ref. [29], copyright *Nat. Commun.* 2020.

**Figure 2 membranes-13-00457-f002:**
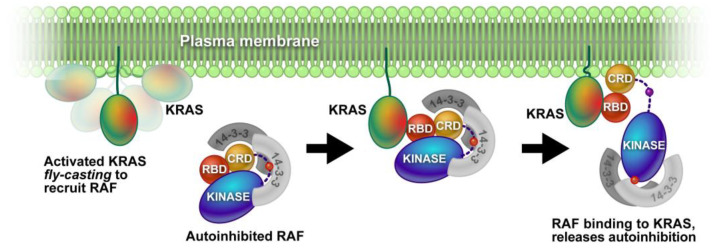
The KRAS G-domain, bound to the bilayer through its HVR, is unseated from the surface with partial conservation of orientation as determined by a membrane contact that is broken in a transient, orientationally defined membrane-bound state. Reproduced with permission from Ref. [43], copyright *PNAS*. 2020.

**Figure 3 membranes-13-00457-f003:**
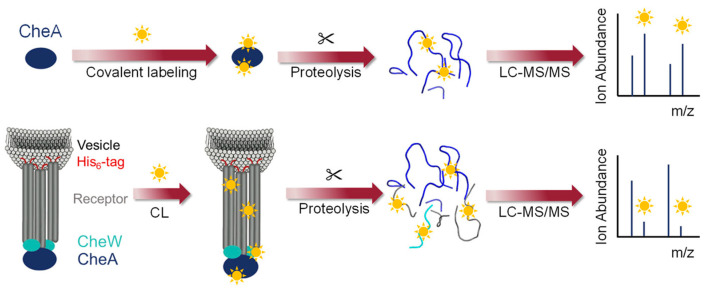
The DEPC CL-MS of bound and unbound CheA. Reproduced with permission from Ref [48]. Copyright *J. Am. Soc. Mass Spectrom.* 2023.

**Figure 4 membranes-13-00457-f004:**
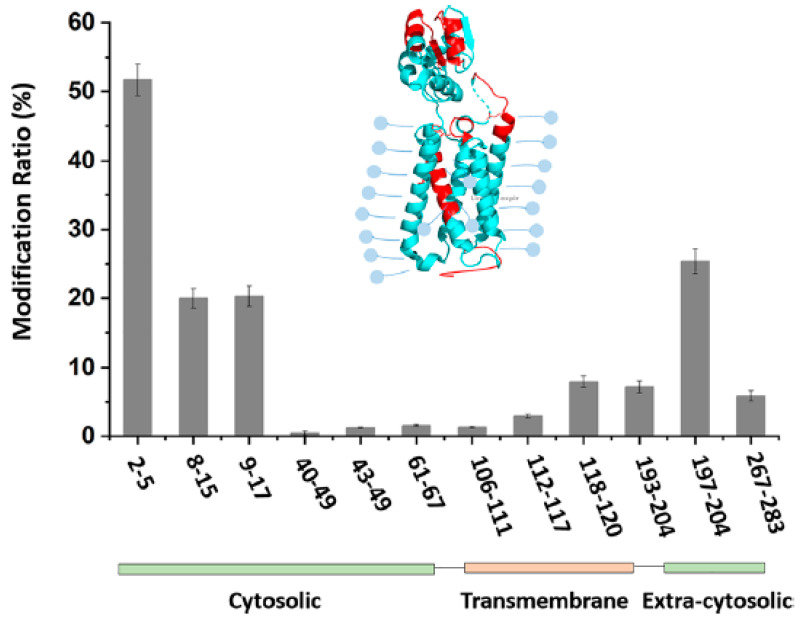
The VKOR was modified by TFBB in a micelle system. The outcome shows the modification ratio at the peptide level. Reproduced with permission from Ref. [57], copyright *Anal. Chem.* 2021.

**Figure 5 membranes-13-00457-f005:**
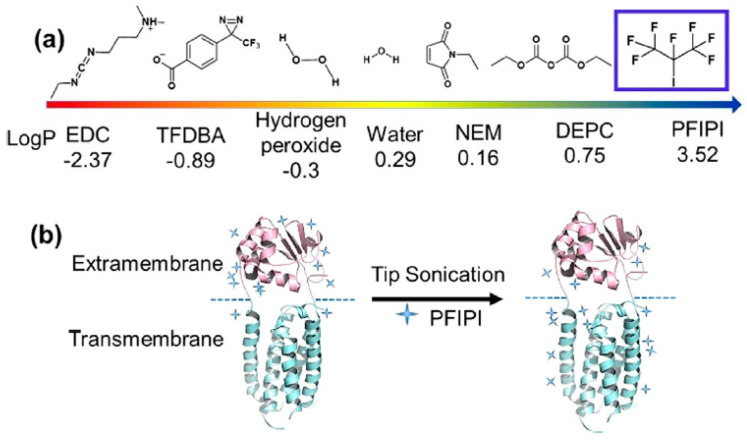
(**a**) The *logP* values of the reagents that were developed for protein transmembrane footprinting. (**b**) Tip sonication treatment accelerates PFIPI penetration allowing the hydrophobic transmembrane region to be footprinted. Reproduced with permission from Ref. [66], copyright *Angew. Chem. Int. Ed.* 2021.

**Figure 6 membranes-13-00457-f006:**
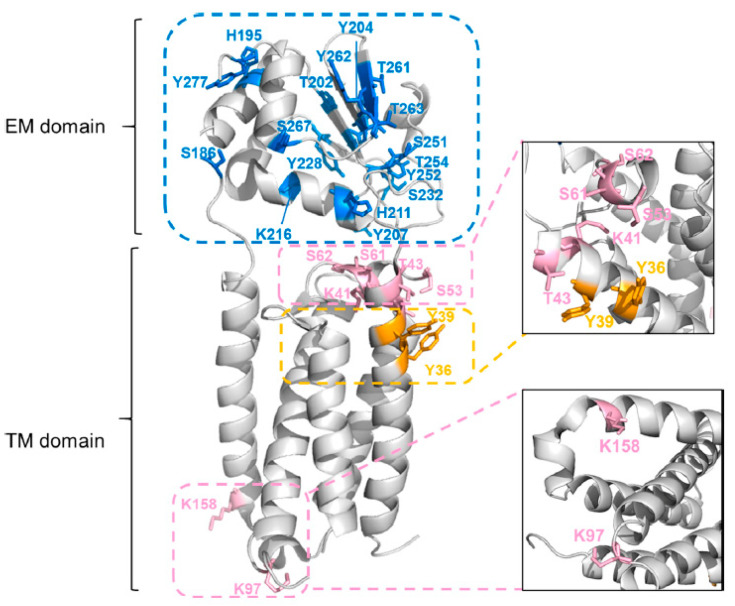
The residues were labeled by DEPC footprinting including those in the EM domain (9 residues) and in the TM domain (21 residues). Reproduced with permission from Ref. [67], copyright *J. Am. Soc. Mass Spectrom.* 2021.

## Data Availability

Not applicable as this is a review with no new data.

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
