# Peer review of "Advances in Mass Spectrometry on Membrane Proteins"

_membranes, 2023, doi:10.3390/membranes13050457_

Round 1

Reviewer 1 Report

This is a nice review that will provide novices with rapid access to recent developments related to MS-based studies of membrane protein structure and dynamics. The manuscript will be acceptable after minor revisions.

Specific Comments:

Please revise the title. The last word (“Introduction”) should be deleted, or the entire title should be reworded.

Please check ref. 15 (incomplete)

It would be better to avoid using possessive apostrophes in the abstract (MPs’, proteins’, etc.).

p. 1: “Recently, MS-based approaches are being used …” check grammar

p. 1/2. “Two major categories of MS- based approaches are structural proteomics and native MS.” This statement is contentious. Several of our colleagues would consider native MS to be part of structural proteomics. See for example: Benesch et al. (Chem. Rev. 2007, 107, 3544−3567), Title: Protein Complexes in the Gas Phase: Technology for Structural Genomics and Proteomics

p. 2: “MPs have both hydrophobic (non-soluble) and hydrophilic (soluble) regions.” Misleading. This is also true for non-membrane globular proteins, all of which have a hydrophobic core. The key point is that MPs have exposed hydrophobic regions on their surface, and these regions are normally covered by the lipid alkyl chains.

p. 2: “as detailed in several articles published in Nature Protocols [13].” A bit awkward to only cite one paper after this statement.

p. 2: “in deuterated solvent” please be more specific (D2O). There are many deuterated solvents.

The last sentence of the abstract, as well as line 181, correctly imply that HDX is not really a ”footprinting” technique. This contradicts the wording on p. 3 : “1-2-1. Footprinting transmembrane domains

Definition of IF and OF acronyms seems to be missing.

No need to define the “HOS” acronym on p. 5, because it is not used again.

p. 7: For non-experts, it would be nice to explain what a picodisc is.

Line 280 “Besides .OH …” The beginning of this paragraph is not well connected to the previous text which discussed NEM footprinting.

Line 373: “Recently new “footprinters” came under development to extend the application of FPOP.” This opening sentence is a bit awkward as well because the previous text already discussed novel FPOP strategies.

I find it difficult to understand Vachet’s Figure 3. Why do both of the labeled peptides show lower intensity in the bottom spectrum? According to the protein cartoon, one of the labeled peptides (the one not covered by the receptor) should retain its original intensity.

Reviewer 2 Report

Yang et al have written a review article which excellently summarizes recent milestones in the use of footprinting MS strategies to understand membrane protein structure-function. The organization into membrane associated, extra membranous and integral membranous sections made it easy to read and follow, as well as provides an adequate place to discuss the limitations, and possible solutions, to achieving useful information in these protein areas.

However, it confusingly states that this review focuses on achievements over the past five years that enable the study of membrane proteins by MS. This review is not comprehensive or broad enough in its current form to make this claim and should be refocused on what it does exceptionally well, which is to provide an informative and interesting account on footprinting MS for membrane protein analysis – and not Native MS, HDX-MS or cross-linking MS. The title and abstract would need to be changed, in particular, as well as areas of the main text to highlight this suggested re-focusing.

Whereas the comparison the HDX-MS is useful (as this is essentially a footprinting technique with deuterium) they do not provide the depth required for a broad readership to gain new knowledge of the HDX-MS membrane protein field and should direct them to more comprehensive reviews. An example of this point is the statement:

1-3. Future directions for HDX section: “The detergent and lipids can interfere with the signal coverage of MPs. These contaminants can cause aggregation and alter the native conformation of the protein, leading to incomplete or inaccurate HDX signals. The development of methods such as detergent-free purification or the use of liposomes to minimize the interference of these contaminants should help.”

There have been advances in these areas over the years which were not mentioned/covered here, e.g.:

DOI: 10.1021/acs.analchem.1c01171       

DOI: 10.1016/j.chemphyslip.2019.02.007

DOI: 10.1021/acs.jproteome.7b00830    

DOI: 10.1021/acs.analchem.2c04876

DOI: 10.1002/anie.201709657

DOI: 10.1021/acs.analchem.8b00429

DOI: 10.1016/S1044-0305(02)00702-X

These could be included to improve the review but, perhaps, these don’t (all) need to be cited or mentioned if the review is refocused as suggested above.

Minor issue(s):

“analysis more complete than for HDX.” P. 5, line 180 – is the analysis more complete than HDX? In what sense? Or is the analysis simply easier to perform/adapt/troubleshoot due to the irreversible nature of the label?

Schmidt et al. [43],[44] used – these Schmidt papers don’t seem to exist in the reference list provided.

Round 2

Reviewer 2 Report

Yang et al have addressed all concerns put forward and the refocused review reads well and will be informative to the field. I warmly recommend the manuscript for publication in Membranes.